# Single-Stage Tibial Osteotomy for Correction of Genu Varum Deformity in Children

**DOI:** 10.3390/children10020377

**Published:** 2023-02-14

**Authors:** Nikolas Kolbe, Frank Haydon, Johannes Kolbe, Thomas Dreher

**Affiliations:** 1Department of Orthopedics and Traumatology, University Hospital Heidelberg, 69118 Heidelberg, Germany; 2Orthopedic Surgeon, NGO Mercy Ships, 1012 Lausanne, Switzerland; 3Head of Pediatric Orthopedic and Trauma Surgery, Children’s University Hospital Zürich, 8032 Zürich, Switzerland; 4Head of Pediatric Orthopedics, Orthopedic University Hospital Balgrist, 8008 Zürich, Switzerland

**Keywords:** pediatric orthopedics, genu varum, bow leg, tibial osteotomy, Blount disease, rickets, growth disorder

## Abstract

Conservative and operative treatments with gradual or acute correction of severe varus deformities of the leg have been described. We evaluated whether the corrective osteotomy used within the NGO Mercy Ships is an effective treatment for genu varum deformity of different etiologies in children and which patient specific factors have an influence on the radiographic outcome. In total, 208 tibial valgisation osteotomies were performed in 124 patients between 2013 and 2017. The patients’ mean age at the time of surgery was 8.4 (2.9 to 16.9 (min/max)) years. Seven radiographically measured angles were used to assess the deformity. Clinical photographs taken pre- and postoperatively were assessed. The mean time between the surgery and the end of physiotherapeutic treatment was 13.5 (7.3 to 28) weeks. Complications were monitored and classified according to the modified Clavien–Dindo-classification system. The mean preoperative mechanical tibiofemoral angle was 42.1° varus (range: 85°–12° varus). The mean postoperative mechanical tibiofemoral angle was 4.3° varus (range: 30° varus–13° valgus). The factors predicting a residual varus deformity were higher age, greater preoperative varus deformity and the diagnosis of Blount disease. The tibiofemoral angle measured on routine clinical photographs correlated well with the radiographic measurements. The single-stage tibial osteotomy described is a simple, safe and cost-effective technique to correct three-dimensional deformities of the tibia. Our study shows very good mean postoperative results, but with a higher variability than in other studies published. Nevertheless, considering the severity of preoperative deformities and the limited opportunities for aftercare, this method is excellent for the correction of varus deformities.

## 1. Introduction

Genu varum deformities of the leg are often associated with internal tibial torsion [1] and can occur unilaterally, bilaterally or as part of a windswept deformity. Amongst the causes for a genu varum deformity are physiological bow legs, Blount disease, rickets, infections, traumatic growth plate injuries, skeletal dysplasias and neoplasms [2,3]. Common causes for a genu varum deformity in Africa are Blount disease and rickets [4,5,6]. However, little is known about the exact incidence of these diseases. One report estimates the prevalence of infantile Blount disease in the Caribbean at 1/1200 live births [7,8], with bilateral involvement in 37–62% [9,10,11,12]. Rickets remains a significant health problem in developing countries [13], with the prevalence in different African countries ranging from 3% to 42% [5]. A population study in The Gambia found the clinical criteria of rickets in 3.3% of children under the age of 18, while only 0.6% showed radiographic signs of rickets. In this study, bilateral bow leg deformity was the most common deformity (53%), followed by knock knee deformity (47%).

If bow leg deformities do not resolve spontaneously or by means of bracing, or exceed a certain severity, a surgical treatment is needed to prevent the progression, and thus the long-term effects of an unevenly distributed weight at the knee, which are gait disturbance, pain, joint instability and osteoarthritis [2,14,15]. Another important aspect to consider is the psychological impact of limb deformities, especially in cultures where diseases are perceived as a punishment for the patients or their families.

Surgical treatment can either involve guided growth [16,17,18] or acute or gradual osteotomy [19,20,21]. Various osteotomies have been described in the literature [22], with various fixation methods including external fixators [23,24,25], screws [19,26] or pins [27].

Around five billion people lack access to safe and affordable surgical and anesthesiologic care when needed, which can be due to structural deficits, or financial, political and geographical obstacles. In order to cover the general surgical need of a population, a minimum average surgical rate of 5000 procedures per 100,000 people per year is necessary. Many low- and middle-income countries do not meet this need, an issue which is most prominent in Sub-Saharan Africa. In these regions, an additional 4343 to 5625 procedures per 100,000 are needed to meet the population’s demand [28]. Children from Sub-Saharan Africa with lower limb deformities as part of the above-mentioned risk population therefore have very little access to safe surgical corrective procedures.

The surgical method to correct three-dimensional tibial deformities described in this publication is used within the non-governmental organization (NGO) Mercy Ships. The purpose of this NGO is to bring free surgical and medical care to countries with deficient health care systems. The facility used is a hospital ship, which stays in one port for about 8–10 months. Due to these special settings, a reliable and immediate corrective procedure for the three-dimensional deformity with relatively few financial expenses is needed. Guided growth techniques are not used, because an adequate follow-up cannot be guaranteed. Some children may only have the opportunity to undergo a safe surgical procedure once, and therefore the surgeries are performed despite the possible risk of recurrence. The purpose of this publication is to evaluate a single-stage proximal tibial osteotomy used by the NGO Mercy Ships to correct genu varum deformities in combination with rotational and sagittal plane deformities according to its potential to acutely correct the deformities and improve posture and patient satisfaction, as well as to report potential complications.

## 2. Materials and Methods

Approval for this retrospective study was granted by the Mercy Ships Institutional Review Board (s. Appendix A). The medical records of all patients who underwent single-stage tibial osteotomy for bow leg deformity on the ship Africa Mercy were reviewed. Patients who received a femoral osteotomy for varus deformity or a tibial osteotomy for valgus deformity were not included. A total of 124 patients from the Republic of Congo, Madagascar, Benin and Cameroon, who were operated on between October 2013 and October 2017, were identified and included. Overall, 84 patients had a bilateral deformity, 25 a unilateral deformity and 15 had a windswept deformity. Of the cases with windswept deformity, only the extremity with the bow leg deformity was included. A total of 208 extremities were studied in 53 males and 71 females. The patients mean age at time of surgery was 8.4 (2.9 to 16.9 (min/max)) years. The mean time between the surgery and the end of physiotherapeutic treatment was 13.5 (7.3 to 28) weeks, with the last radiographic control being at 13.4 (5.9 to 28) weeks postoperatively. Due to the Ebola outbreak of 2015, the NGO mercy ships unexpectedly stayed in one country for two consecutive years. All patients of the previous year, who were treated with a tibial osteotomy for varus deformity, were invited for a follow-up examination. Of the 13 patients invited, a short-term clinical and radiographic follow-up one year postoperatively was conducted in 5 patients (9 limbs).

For measurements of the radiographic parameters, weight-bearing long leg anteroposterior studies were used. These were taken preoperatively and at least at two follow-up examinations: one at 1-2 weeks in the cast to check for alignment and possible need of cast wedging, and at around 10–14 weeks, just before the patients were discharged. If wedging was necessary, an additional long leg radiograph was taken to assess the alignment after wedging. Regular AP and lateral X-rays were taken to check for bony consolidation at 6–8 weeks. The following 7 angles were used to assess the radiographic deformity [29,30]: mechanic lateral distal femoral angle (mLDFA), mechanic medial proximal tibial angle (mMPTA), joint line convergence angle (JLCA), anatomic and mechanic tibiofemoral angle (aTFA, mTFA), tibial metaphyseal diaphyseal angle (MDA) and tibial plateau-angle. To evaluate the treatment, the difference between the last preoperative measurement and the last postoperative measurement after bony consolidation was determined. According to Smith et al. [31], the radiographic outcome was graded into “excellent” with a postoperative mechanical axis within -6 to 3 degrees, as “fair” with a mechanical axis of −11 to −7 or 4 to 8 degrees and as “poor” with a mechanical axis outside of this range.

As a part of the standardized evaluation program on the Mercy ship, pre- and postoperative photographs from the front and the back were taken for all patients. The tibiofemoral angle, as the angle between the thigh and the lower leg, was measured on those photographs, and a mean between the photograph from the front and the back was taken for evaluation. The lack of conventional follow-up possibilities created the idea of an unconventional follow-up with clinical photographs. The accuracy of photographic measurements of the tibiofemoral angle was assessed and compared to radiographic measurements of the same angle.

To assess whether photographic documentation allows the approximate detection of changes in bony alignment and can therefore be used to do a further follow-up of the patients, the differences of radiographic alignment and alignment measured on photographs were compared using linear regression models.

Routinely, the tibiofemoral angle and the thigh-foot angle were measured with a goniometer. The range of motion (ROM) was documented according to the neutral zero method by one experienced examiner. A questionnaire (s. Appendix B) was completed to document patient satisfaction. The functional outcome was rated by experienced physiotherapists. The pain score was surveyed on a Numeric Pain Rating Scale, and the functional level rated on a scale from 0–5. A good outcome was defined as improvement either in the functional level or in the pain score. If both parameters stayed the same, the outcome was fair, and if either the functional level decreased or the pain score increased, the outcome was rated as poor.

During our surgical procedure, we followed the subsequent steps (Figure 1). First, the important landmarks at the knee are marked, including the borders of the patella, the joint line and the fibular head (1A). Then, a fibular osteotomy is performed using a vertical incision below the fibular head, making sure not to damage the peroneal nerve. To allow the free movement of the distal leg, especially in patients with severe varus deformity, the resected part of the fibula needs to be about 2–3 cm (1B). Then, an anterior fasciotomy using the same incision is performed.

The tibial incision is made perpendicular to the deformed tibia, located 1 cm below the tibial tubercle (1C). After exposing the tibia and protecting the posterior neurovascular structures with retractors, the subperiosteal osteotomy is carried out using multiple drill passes and completed with an osteotome (1D). The osteotomy is made distal to the tibial tuberosity and perpendicular to the distal part of the deformed tibia.

In 90° flexion of the knee, the foot is then rotated and translated into the right position, aiming for 10° valgus and approximately 10–15° external rotation due to the tendency of recurrence of the deformity [32,33]. This step is important for the planning of the removal of a bony notch (1E).

The next step is to remove a notch from the distal fragment of the tibia. The notch has to match the proximal tibial osteotomy fragment. Therefore, the more severe the deformity, the more medial the apex of the notch is positioned. If the desired correction is exactly 45°, then the apex of the notch has to be in the middle of the tibial shaft (1F). First, the apex of the notch is marked with a drill pass, then it is completed using a Luer or osteotome (1G). The fragments have to “click” together (1H) and are then fixed with a single screw or a pin (1I). The varus deformity and the rotational deformity need to be considered by positioning the distal leg into the corrected position before removing the notch. The procurvatum/recurvatum deformity can be corrected by angulating the osteotomy in the notch in the sagittal plane. In addition to internal fixation, a long leg cast is applied in a 10° valgus position and, if present, the pin site is windowed to allow pin care on day 5, 8 and 11. The cast is split immediately to accommodate the swelling.

After overwrapping the cast, weight-bearing usually starts on day 4 to 5 with walkers or crutches, achieving full weight-bearing on day 8 to 11. The pin is removed on day 11, and weight-bearing anteroposterior radiographs are taken. If the alignment is not satisfactory, the cast can be wedged. The aim is to push the mechanical axis to the middle of the lateral femoral condyle. The wedging is done after removal of the pin at the height of the osteotomy and controlled with another radiograph. The cast is changed after 3 weeks and removed after 6 weeks. Radiographs are taken to assess the healing process and, if needed, the leg is re-casted for another 2–4 weeks.

If the healing is sufficient, intense physiotherapy training is performed 2 to 3 times a week for several weeks. If satisfying muscle strength and ROM are achieved, the patients are discharged.

All patients are given daily dietary supplements, including vitamin D, calcium and other vitamins during their treatment and for several months after discharge. However, adherence to this drug therapy cannot be controlled. Furthermore, parents are educated on alimentary diversity in their children’s diets.

Despite establishing cooperation with local medical facilities and training medical staff, obstacles such as the lack of health insurance coverage, long travelling distances and low compliance prevent routine follow-up appointments with local medical staff.

The following clinical example (Figure 2) shows a 7-year-old girl with Blount disease, treated with the above-mentioned procedure.

Complications were monitored and classified according to the modified Clavien–Dindo-classification system for unintended surgical events [34].

For statistical analysis, the program IBM SPSS Statistics 23 was used. Normality test was performed on all collected parameters using the Shapiro–Wilk test. Normally distributed data were analyzed with the Paired Sample T-Test, while non-normally distributed data and samples of n < 30 were analyzed using the Wilcoxon test. When comparing more than two independent groups, the Kruskal–Wallis test was used.

Correlations were tested using linear regression models. The statistical significance was set at *p* < 0.05. For all patients with bilateral osteotomies, we randomly selected one leg in order to rule out patient-specific factors for the outcome of both legs.

## 3. Results

### 3.1. Clinical Results

Out of the 124 patients, 60 patients (48.4%) were diagnosed with Blount disease, 45 (36.3%) with rickets, 12 (9.7%) with idiopathic genu varum and 7 (5.6%) with diseases of another etiology. Fixation of the osteotomy was achieved with Steinmann pins in 100 patients (80.6%) and with a single cannulated screw in 22 patients (17.7%). One patient received no fixation, and for another one, the method was not documented. The pins were pulled after an average of 9.5 (1 to 21) days. The mean time for casting was 8.6 (4.6 to 23.6) weeks. Older children required a longer treatment with casts until bony consolidation was completed (1.8 days/year, *p* < 0.001).

A reduction of the clinically measured TFA from an average of 35.7° varus (range: 85° varus–5° varus, SD 15.5°) to 2.0° valgus (range: 16° varus–15° valgus, SD 7.0°) was achieved (*p* < 0.001) (Table 1). The thigh-foot-angle changed from a mean of 39.1° IR (internal rotation) (range: 110° IR–17° ER, SD 22.5°) to 3.1° IR (range: 50° IR–40° ER, SD 15.3°) (*p* < 0.001). Preoperatively, 20 patients showed a genu recurvatum, defined as an extension in the knee greater than 10°. The average recurvatum was 18.8° (range: 12°–40°, SD 7.0°). Additionally, 12 patients showed an average 12.6° (range: 5°–25°, SD 6.9°) knee extension deficit. Postoperatively, two patients showed an average 20° recurvatum (range: 15°–25°, SD 7.1°). Both patients showed a normal knee extension preoperatively. A postoperative knee extension deficit was noted in 12 patients with an average of 6.1° (range: 3°–11°, SD 2.6°). Five of those twelve patients showed a preoperative knee extension deficit between 10° and 20°, and thus all improved through the surgery, whereas seven patients had normal preoperative knee extensions.

The questionnaire was fully completed in 93 out of 124 patients. Overall, 69 (74.2%) patients achieved a good, 20 (21.5%) a fair and 4 (4.3%) a poor outcome. One of the patients with a poor outcome was discharged early against medical advice because of personal reasons of the caregiver, and another patient had postoperative bleeding, which required a blood transfusion of one unit and a reintervention with ligation of the medial superior genicular artery. The other two had minor complications related to the cast.

There was a significant association between an increase in the functional score as assessed by the physiotherapists (s. Appendix C) and a decrease in the pain score (*p* = 0.015). The greater the preoperative deformity, the greater the pain score (0.046/degree, *p* = 0.002) and the worse the level of function (−0.013/degree, *p* = 0.005).

Older patients had a worse level of function (−0.064/year, *p* < 0.001) and a higher preoperative pain score (0.263/year, *p* < 0.001), having also a higher decrease in the pain score through the surgery (0.238/year, *p* < 0.001). Furthermore, there was an association between the preoperative mechanical axis and the decrease in the pain score (−0.045/degree, *p* = 0.002). There were no significant associations between the preoperative mechanical axis and changes in the functional level.

### 3.2. Radiographic Outcomes

The radiographic parameters are shown in Table 1. The mLDFA, the mMPFA, the MDA, the aTFA and the mTFA were improved on a significance level of *p* < 0.001. The JLCA and the plateau-angle did not change significantly.

According to our radiographic grading based on the postoperative mechanical tibiofemoral angle, an excellent outcome was achieved in 48 out of 118 patients (40.7%), 39 patients achieved a fair radiographic outcome (33.1%) and 31 patients achieved a poor radiographic outcome (26.3%).

Of the 31 patients with poor radiographic outcome, six were due to overcorrection with an average mechanical tibiofemoral angle of 10.5° valgus postoperatively (range: 9° valgus–13° valgus).

The other 25 patients had a mean age of 10.3 years (range: 3.8–16.9 years) and were thus above the general mean age. Furthermore, their average preoperative deformity of 56.0° varus (range: 85° varus–23° varus) was greater than the general mean. Additionally, 15 of those 25 patients (60%) were diagnosed with Blount disease. These are all factors that contributed to a poorer outcome. The average correction in the mechanical axis through the surgery was 38.9° (range: 8°–69°). A total of eight complications was noted in the above mentioned 25 patients (32% complication rate in those with poor radiographic outcome due to residual varus deformity). Two complications were major complications. One patient showed vascular compromise requiring revision surgery with a lower correction angle, new placement of the pins and a cast change, and the other a cast failure and excessive postoperative pain requiring a cast change in the operating room.

Various influencing factors for the postoperative mechanical axis were identified (Table 2). The factors that caused more varus of the postoperative mechanical axis were higher age, more severe preoperative deformity, an insufficiently overcorrected mechanical axis in the cast and the diagnosis Blount disease.

The importance of the diagnosis for the postoperative result is emphasized in Figure 3. Of the patients with all the necessary x-rays available, 56 patients were diagnosed with Blount disease, 44 with rickets and 10 with idiopathic genua vara. The groups of different etiologies do not significantly differ in preoperative mTFA nor in postoperative mTFA in the cast, while the difference is significant in their mean age (Blount disease: 9.0 years, Rickets: 6.8 years and Idiopathic: 8.1 years, *p* = 0.003) and the loss of correction until bony consolidation, and thus the mTFA at discharge (*p* < 0.05).

A short-term radiographic follow-up one year postoperatively was available in five patients (nine limbs). Because of the small sample, we display all nine limbs assessed. Patients reported a high satisfaction with the surgery at follow-up. The average age at the time of surgery of this sample was 6.4 years old and, thus younger than the overall average age. All five patients were diagnosed with rickets. Table 3 shows the radiographic results in these nine limbs preoperatively, for the time of discharge and the time of follow-up. The mean mechanical alignment moves from an almost neutral axis at the time of discharge to a slight valgus alignment at the time of follow-up with a mean change in mechanical axis from 1.3° varus to 4° valgus.

### 3.3. Photographic Measurements

We obtained pre- and postoperative photographs of 111 patients. The mean difference between the preoperative axis and the postoperative axis measured on the photographs was 44.1° (range: 5°–86.5°, SD 17.3°) (Table 1). Adjusted for age, there is a strong association between the photographic TFA and the radiographically measured aTFA (0.84; *p* < 0.001), and between the photographic TFA and mTFA (0.70; *p* < 0.001) (Table 4 and Figure 4). The mean difference between the radiographically and photographically measured angles is 7.0° for the aTFA and 8.1° for the mTFA.

### 3.4. Complications

An overall complication rate of 23% was detected. Overall, 24 of 124 (19%) were minor complications, and five of one-hundred twenty-four (4%) major complications. Out of the twenty-four complications documented, three cases were related to a general surgical intervention, ten cases were related to the specific surgical intervention and the post-treatment, five cases were related to the surgical wound, ten cases were related to the cast and three cases were due to a general infection not associated with the surgical wound. For further details, see Table 5.

## 4. Discussion

In societies with other religious beliefs, deformities have, besides their medical aspect, a great cultural influence on the patients and their families, which may complicate social participation [35]. Therefore, the main motivation of the patients and their caregivers to seek medical treatment is often to restore a normal visual appearance and end social stigmatization.

The hereby described surgical technique allows for the correction of malalignment in the coronal and sagittal plane as well as the correction of rotational deformities. It is a simple and cost-effective method, which can be used for different entities and severities of genu varum deformities. It allows reproducible results with a low complication rate. With the help of this technique, even severe deformities can be corrected with a single surgical intervention. The early pin pulls and transition to full weight-bearing reduces pin infections and allows for a relatively fast recovery of the patients. This is especially important in the setting of the NGO Mercy Ships, where the time frame for treating patients is limited and the local medical system does not provide the possibility of a reliable follow-up. Therefore, the patients are only discharged when the postoperative physiotherapeutic treatment provided by Mercy Ships is completed and patients are self-dependent.

Many different techniques have been described in the literature to correct genu varum deformities. The radiographic results are often presented in various ways, which makes comparison difficult. Comparing to other published results for acute proximal tibial valgisation osteotomies without the use of external fixation, the results revealed that with a number of cases of n = 124, our cohort was one of the biggest studied [36,37,38,39,40,41,42,43,44]. The majority of authors studied the surgical technique in just one etiology of the deformity, whereas Dilawaiz et al. [44] published the results for a focal dome osteotomy used for varus and valgus deformities caused by skeletal dysplasia, tibia vara and metabolic bone disease. Our cohort also included patients with Blount disease, rickets, idiopathic genu varum, skeletal dysplasia and post-traumatic deformity.

In our study, the mean preoperative anatomic tibiofemoral angle was 42.9° varus (range: 89° varus–6° varus) and, thus more severe than the mean preoperative anatomic tibiofemoral angles of 35° varus to 16.6° varus treated with comparable techniques [36,37,38,39,40,41,42,43,44].

Postoperatively, our cohort showed an anatomic tibiofemoral angle of 2.3° valgus (range: 21° varus–24° valgus). Dilawaiz et al. [44], who treated different etiologies, as mentioned above, reported a postoperative anatomical tibiofemoral angle of 2° varus (range: 5° varus–2° valgus). Though the mean postoperative outcome is comparable to our results, the range in our study is much wider, and thus the postoperative outcome is less reliable. However, with their mean preoperative anatomic tibiofemoral angle of 18° varus (range: 28°–6° varus), the deformity was less severe than in our cohort.

Furthermore, our data show a variable outcome, depending on the underlying pathology (s. Figure 3). The differences between the mechanical alignment at discharge of the Blount disease group and the other two groups are significant. This is due to several reasons.

In patients with severe joint deformity, especially with advanced stage Blount disease, in addition to the bony deformity, the laxity of the lateral ligamentous structures and, in some cases, a subluxation in the frontal plane contribute to the deformity. In the first postoperative x-ray image after 8 to 11 days, the patients’ knees are stabilized in a long leg cast, and therefore ligamentous laxity is not relevant for the alignment. The discharge X-rays are done without a cast or brace in a standing position, in which the laxity of the ligaments, if present, will contribute to a varus alignment.

Furthermore, our single stage correction does not address intra-articular deformity; therefore, in patients, especially with advanced stage Blount disease, the depression of the medial tibial plateau creates a varus thrust [4], and thus favors recurrence.

The age of the patients differs significantly between the groups. The average age of patients with Blount disease was 9.0 years, while the average age for the rickets group was 6.8 years and for idiopathic, 8.1 years. Older children in our study required a longer cast treatment, and recurrence was favored.

Wesselsky et al. [37] reported on a cohort of patients with rickets treated by a NGO with a mean preoperative anatomic tibiofemoral angle of 31.2° varus (range: 51°–14° varus; SD = 11.8°). Of the 15 patients treated for a varus deformity, only nine were treated with a tibial osteotomy and the other six with a femoral osteotomy. Their mean postoperative anatomic tibiofemoral angle was 0.3° valgus (9° varus–7° valgus). In our cohort, the mean postoperative anatomic tibiofemoral angle for the subgroup “rickets” was 5.1° valgus (range: 21° varus–20° valgus, SD 9.6°).

Using the criteria of Smith et al. [31], we only achieved a desirable excellent outcome in 41% and a poor outcome in 26% of our patients. We already stated that various factors contributed to a poorer outcome in these patients. Our cohort presented greater preoperative deformity, greater age and more frequently had Blount disease as an underlying pathology. In contrast to our study, most published results and techniques are performed in settings with sufficient economic and structural means [22,24,31,45,46,47,48,49].

Our results show a good approximation of the measured knee angles towards the physiologic angles and are comparable to other published radiographic results. The range measured in the postoperative anatomic and mechanical tibiofemoral angles is, however, wide and leaves room for improvement. Especially in older children, a more precise outcome could be achieved by using gradual correction with an external fixation [20]. However, considering the limited economic and time resources given, partial correction of the deformity should also be considered a success, with many positive impacts for the patients.

The limitations of our study are the setting and the resources available for treatment. Through the screening process, a heterogenic group of patients was generated with different etiologies of bow leg deformity. The diagnosis was mainly based on the radiographic evaluation, and there was no full blood workup done routinely to detect, e.g., different forms of rickets.

The results of our study are mainly based on radiographic evaluation. However, especially preoperatively with often severe and three-dimensional deformities, it is sometimes difficult to take a standard AP radiograph. If the rotation of the foot cannot be adjusted correctly, the angle measurements in the frontal plane will be altered.

Another limitation is the lack of complete clinical and radiographic data in all patients. For our main endpoint, which is improvement in mechanical alignment of the leg, there were 116 of 124 cases in which all necessary x-rays were present.

The etiologies of the deformity, especially Blount disease, have a high rate of recurrence per se. Authors reported recurrence rates of up to 94% [32,33]. In patients with rickets, a recurrence rate of up to 90% is reported [50,51]. This will compromise the long-term outcome of our patients treated, especially because there is seldom the possibility to perform a second corrective procedure.

Another important limitation of our study is the lack of long-term follow-up, which is owed to the setting of the work done by the NGO Mercy Ships. We therefore do not know the long-term results of our described method and can only anticipate these with the help of other studies published. In order to improve this, a possible solution would be to invite the patients to take standardized photographs of themselves, e.g., every 6 months. Brooks et al. [52] already recommended the use of photographs to detect the progression of physiologic bow legs. Figure 4 shows an excellent correlation between the mechanical/anatomical tibiofemoral angle and the tibiofemoral angle measured on photographs with little outliers. It is important to take the mean difference of radiographically and photographically measured angles into account. However, especially with the help of several photographs over the course of time, one can very well detect the possible recurrence of the deformity and then coordinate further treatment.

## 5. Conclusions

In conclusion, this study shows good results of proximal tibial osteotomies for three-dimensional deformities with a simple and cost-effective method. In these patients, there is a good correlation between the radiographically and photographically measured long leg angles, which provides the possibility for further follow-up examinations via standardized patient photographs.

## Figures and Tables

**Figure 1 children-10-00377-f001:**
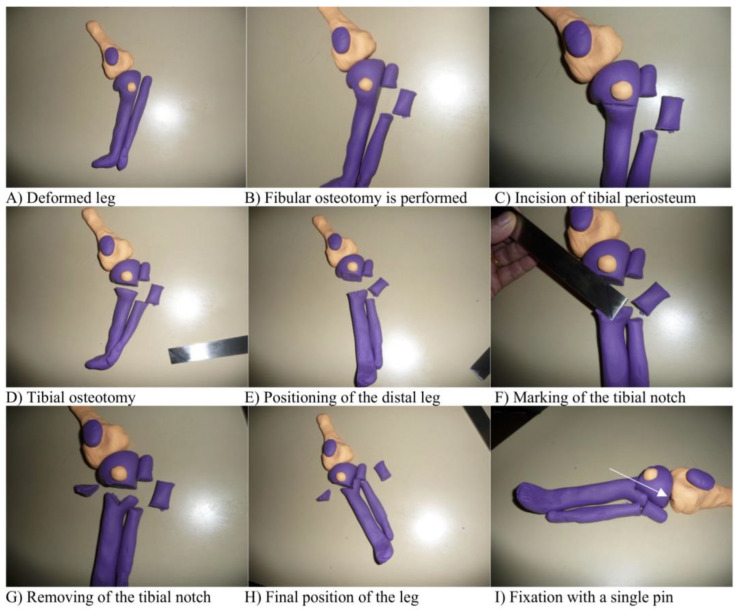
(**A**–**I**). Model of surgical steps with purple dough representing the tibia and the patella and the light dough representing the femur and the tibial tuberosity.

**Figure 2 children-10-00377-f002:**
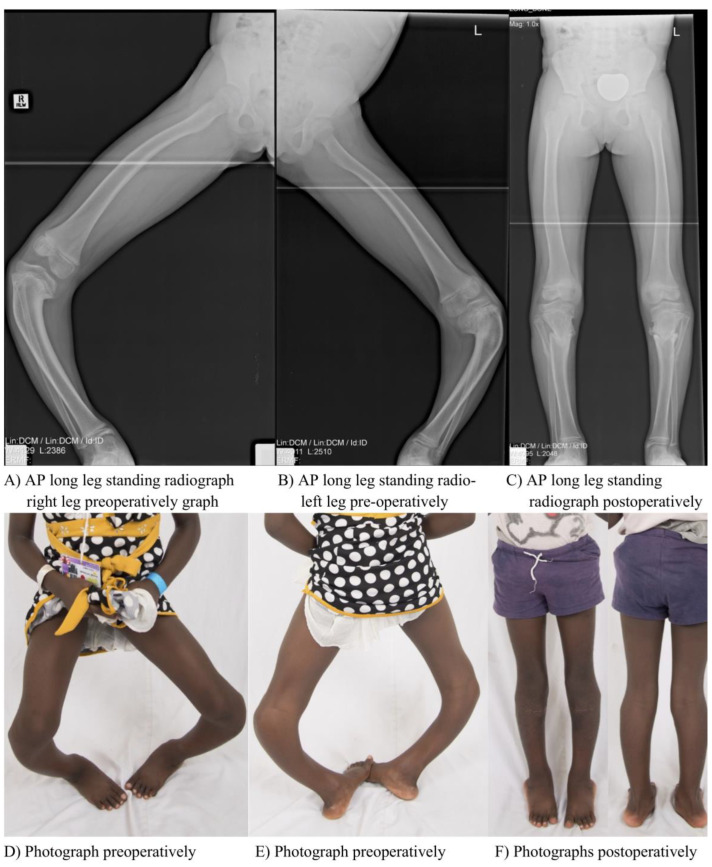
(**A**–**F**) Clinical example with AP long leg standing radiographs and photographs pre- and postoperatively.

**Figure 3 children-10-00377-f003:**
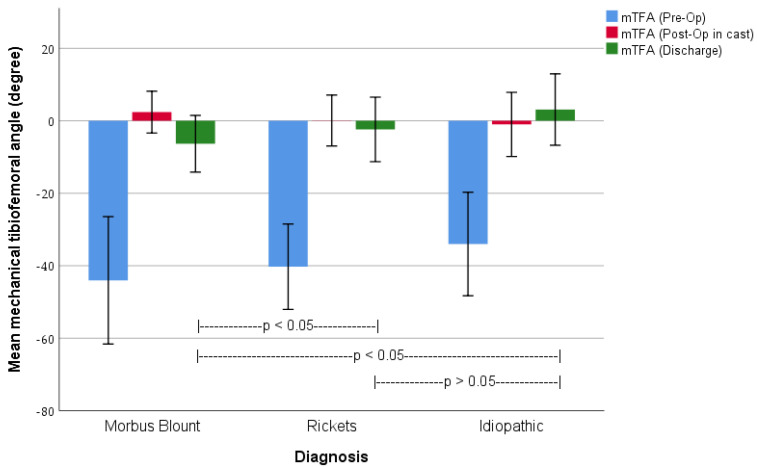
Chronological development of the mean mechanical tibiofemoral angle grouped by etiology of the deformity. Boxes represent the mean value, whiskers show the standard deviation, and *p*-values show significances between the different etiologies of the mean value of mTFA at discharge.

**Figure 4 children-10-00377-f004:**
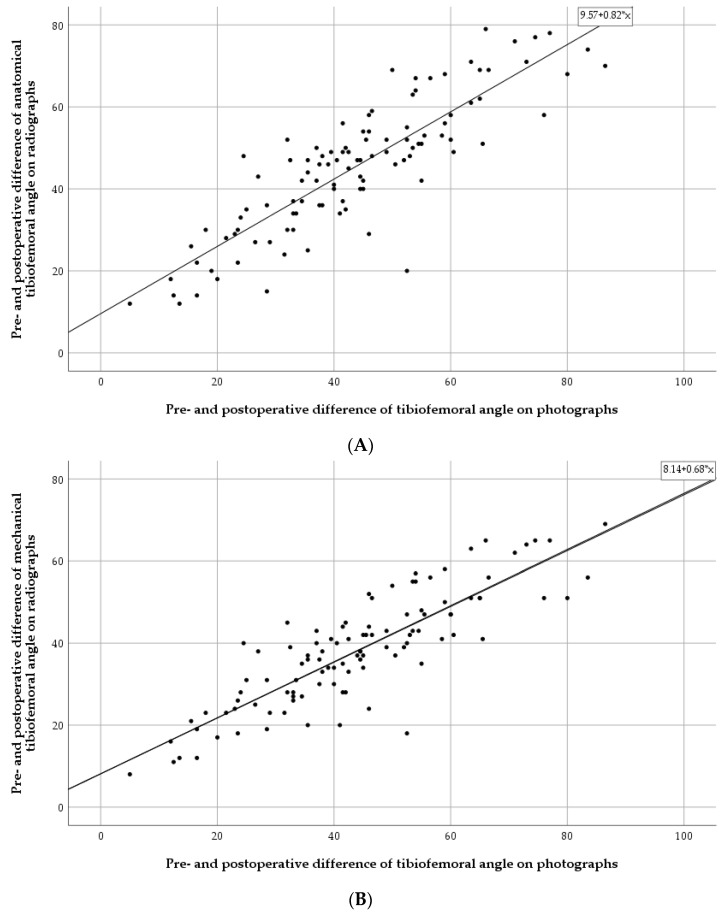
Scatter plots of pre- and postoperative differences of the tibiofemoral angle measured on radiographs and photographs. (**A**) Anatomical tibiofemoral angle and photographs. (**B**) Mechanical tibiofemoral angle and photographs.

**Table 1 children-10-00377-t001:** Results of radiographic, clinical and photographic measurements. N states the number of patients with completed measurements. T1 refers to the preoperative measurements. T2 refers to the measurements taken directly postoperative in a cast. T3 refers to the measurements taken at discharge.

Measurement	n	T1 ^1^ Mean	Min	Max	SD	T2 ^2^ Mean	Min	Max	SD	T3 ^3^ Mean	Min	Max	SD
mLDFA	106	99.8	60	137	14.8	95.3	61	126	13.8	97.8 (*p* < 0.001)	68	130	12.3
mMPTA	106	57.8	4	89	20.7	92.7	66	119	13.7	90.6 (*p* < 0.001)	54	120	14.8
JLCA	106	6.6	0	26	5.6	5.8	0	27	4.7	5.1 (*p* > 0.05)	0	17	3.4
MDA	87	26.9	0	75	16.0	−7.2	−33	48	14.7	−5.1 (*p* < 0.001)	−30	67	16.4
Plateau-angle	102	38.5	2	74	15.7	38.3	1	74	16.8	37.6 (*p* > 0.05)	3	70	16.6
aTFA	117	−42.9	−89	−6	17.4	8.9	−20	31	8.0	2.3 (*p* < 0.001)	−21	24	9.8
mTFA	116	−42.1	−85	−12	15.5	1.2	−20	17	6.6	−4.3 (*p* < 0.001)	−30	13	8.9
Clinical TFA	109	−35.7	−85	−5	15.5					2.0 (*p* < 0.001)	−16	15	7.0
Thigh-Foot-Angle	83	39.1 IR	110 IR	17 ER	22.5					3.1 IR (*p* < 0.001)	50 IR	40 ER	15.3
TFA Photos	111	−44.6	−100.5	−8	19.8					−0.5 (*p* < 0.001)	−55	20	11.2

^1 ^Pre-operative, ^2^ first post-operative X-ray in cast, ^3^ at discharge.

**Table 2 children-10-00377-t002:** Influencing factors on the mechanical tibiofemoral angle at discharge. Various possible factors were tested for their crude and their adjusted influence. Factors adjusted for are listed with the superscript numbers.

Influencing Factor	Unit	n	Crude Difference (95% CI)	Adjusted Difference (95% CI)	Adjusted Difference (95% CI)
Age		115	*p* = 0.002	*p* = 0.024 ^1^	
	per year		−0.63 (−1.03; −0.24)	−0.42 ^1^ (−0.78; −0.06)	
mTFA (preoperative)		115	*p* < 0.001	*p* < 0.001 ^2^	
	per degree		0.30 (0.20; 0.39)	0.28^2^ (0.18; 0.37)	
mTFA2 (in cast)		115	*p* < 0.001	*p* < 0.001 ^2^	
	per degree		0.47 (0.23; 0.70)	0.46 ^2^ (0.23; 0.68)	
Diagnosis					
	Blount disease	57	*p* < 0.001	*p* = 0.071 ^2^	*p* = 0.035 ^3^
			−6.42 (−8.54; −4.30)	−4.0 ^2^ (−8.37; 0.35)	5.67 ^3^ (0.40; 10.93)
	Rickets	29	*p* = 0.011	*p* = 0.040 ^2^	0.036 ^3^
			4.77 (1.11; 8.42)	4.02 ^2^ (0.18; 7.86)	3.69 ^3^ (0.25; 7.13)
	Idiopathic	20	*p* < 0.001	*p* = 0.001 ^2^	*p* = 0.005 ^3^
			8.37 (4.21; 12.53)	7.60 ^2^ (3.22; 11.98)	5.72 ^3^ (1.74; 9.70)
	Other	12	*p* = 0.101	*p* = 0.195 ^2^	*p* = 0.327 ^3^
			−4.25 (−9.33; 0.84)	−3.43 ^2^ (−8.65; 1.78)	−2.34 ^3^ (−7.04; 2.37)

^1^ Adjusted for mTFA, ^2^ adjusted for age, ^3^ adjusted for age and mTFA.

**Table 3 children-10-00377-t003:** Radiographic results of the short-term follow-up in 5 patients (9 limbs).

Measurement	N	T1 ^1^ Mean	Min	Max	SD	T3 ^3^ Mean	Min	Max	SD	T4 ^4^ Mean	Min	Max	SD
mLDFA	9	112	99	125	8.0	107.1	95	121	8.3	102.1	89	123	11.1
mMPTA	9	72.4	65	82	5.3	102.1	94	117	6.8	103.1	89	121	8.9
JLCA	9	2.8	0	6	2.3	5.8	0	10	3.6	4.4	0	8	2.8
MDA	7	17.9	4	25	7.2	−15.4	−24	−7	5.6	−15.4	−22	−6	5.8
Plateau-angle	9	38	0	59	19.0	34.2	1	52	19.4	32.0	3	53	18.0
aTFA	9	−39.4	−53	−12	12.5	5.2	−7	17	8.7	10.6	−6	26	11.9
mTFA	9	−39.7	−52	−16	11.0	−1.3	−14	10	8.0	4	−11	19	10.6

^1^ Pre-operative, ^3^ at discharge, ^4^ at time of follow-up.

**Table 4 children-10-00377-t004:** Correlations between photographic measurements and radiographic measurements as crude correlation and adjusted for age.

Influencing Radiological Measurement	Crude Difference (95% CI)	Adjusted Difference (95% CI)
	*p* < 0.001	*p* < 0.001 ^1^
aTFA	0.82 (0.73; 0.92)	0.84 ^1^ (0.74; 0.93)
	*p* < 0.001	*p* < 0.001 ^1^
mTFA	0.68 (0.60; 0.77)	0.70^1^ (0.61; 0.78)

^1^ Adjusted for age.

**Table 5 children-10-00377-t005:** List of complications according to the classification of Clavien–Dindo.

Grade	N	Listed Complications
Overall	29 (23.4%)	
1	24 (19.4%)	7 patients with damaged skin integrity due to cast requiring local wound treatment
		1 patient with damaged skin integrity due to cast requiring local wound treatment and antibiotics
		2 patients with skin molds inside a cast
		1 patient with a wound infection requiring local wound treatment
		2 patients with a wound infection requiring local wound treatment and antibiotics
		1 patient with a wound break down requiring local wound treatment
		1 patient with excessive postoperative pain requiring opioids
		2 patients with postoperative urinary retention requiring catheterisation and antibiotics
		2 patients with a pin migration
		3 patients with feverish infections not associated to the surgical wound requiring antibiotics
		2 patients with temporary nerve irritations not requiring treatment
		2 patients with a fall after cast removal, one requiring no treatment and one requiring recasting for 4 weeks
1d	1 (0.8%)	1 patient with a permanent peroneus lesion after ablation of the proximal fibular epiphysis requiring an ankle foot orthosis
3	4 (3.2%)	1 patient with a wound infection requiring antibiotics and revision surgery
		1 patient with a cast failure and excessive postoperative pain requiring a cast change in the OR
		1 patient with a loss of reduction requiring a cast change, pin removal and a wound cleaning in the OR
		1 patient with a vascular compromise requiring revision surgery with less correction angle, new placement of the pins and a cast change

## Data Availability

Research data will be provided by the first author Nikolas Kolbe upon request.

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
