# Peer review of "Single-Stage Tibial Osteotomy for Correction of Genu Varum Deformity in Children"

_children, 2023, doi:10.3390/children10020377_

Round 1

Reviewer 1 Report

The paper is excellent in demonstrating a deformity affecting quality of life and outreach to children in need along with consideration of outcome based on follow up. The demonstration to produce results in one stage and with beneficial results are remarkable and helpful for audience to understand. 

Reviewer 2 Report

1.too short follow up- at least untill end of growth: recurrences.

2.x ray of follow up only to 111 pt

3. more emphasize needed to each of the 3 major groups of patients

Reviewer 3 Report

Dear Authors, 

thank you very much for the very nice presentation of the fascinating research data. 

1) Please add in the methods section which method was used to test for normal distribution. Are all variables normally distributed? If variables are not normally distributed, what testing methods were used?

2) The text primarily uses mean and standard deviation, while the figure 3 then uses median and interquartile range, minimum and maximum. Is it possible to keep this consistent?

4) Can significances still be added to figure 3?

5) Why are the angles more corrected in the discharge control in ideopathic varus than in the postoperative control, while there was a recurrence of deterioration in the other forms (M. Blount, Rickets)? 

6) How were the patients followed up (medication, local surgical connection). Has drug therapy for rickets been initiated to minimize negative influences of the underlying disease on the outcome of surgery? 

7) Are there at least some long-term reports?

8) In Table 2, the German words still need to be adapted into English  (Rachitis, ideopathisch).

Thank you for the great work! 

Round 2

Reviewer 2 Report

now suitable for print